# Disturbed Cardiac Metabolism Triggers Atrial Arrhythmogenesis in Diabetes Mellitus: Energy Substrate Alternate as a Potential Therapeutic Intervention

**DOI:** 10.3390/cells11182915

**Published:** 2022-09-18

**Authors:** Baigalmaa Lkhagva, Ting-Wei Lee, Yung-Kuo Lin, Yao-Chang Chen, Cheng-Chih Chung, Satoshi Higa, Yi-Jen Chen

**Affiliations:** 1Graduate Institute of Clinical Medicine, College of Medicine, Taipei Medical University, Taipei 11031, Taiwan; 2Division of Endocrinology and Metabolism, Department of Internal Medicine, School of Medicine, College of Medicine, Taipei Medical University, Taipei 11031, Taiwan; 3Division of Endocrinology and Metabolism, Department of Internal Medicine, Wan Fang Hospital, Taipei Medical University, Taipei 11696, Taiwan; 4Division of Cardiology, Department of Internal Medicine, School of Medicine, College of Medicine, Taipei Medical University, Taipei 11031, Taiwan; 5Department of Biomedical Engineering, National Defense Medical Center, Taipei 11490, Taiwan; 6Cardiac Electrophysiology and Pacing Laboratory, Division of Cardiovascular Medicine, Makiminato Central Hospital, Okinawa 901-2131, Japan; 7Cardiovascular Research Center, Wan-Fang Hospital, Taipei Medical University, Taipei 11696, Taiwan

**Keywords:** atrial arrhythmogenesis, mitochondria, energy metabolism, glucose oxidation, fatty acid oxidation, ketone body

## Abstract

Atrial fibrillation (AF) is the most common type of sustained arrhythmia in diabetes mellitus (DM). Its morbidity and mortality rates are high, and its prevalence will increase as the population ages. Despite expanding knowledge on the pathophysiological mechanisms of AF, current pharmacological interventions remain unsatisfactory; therefore, novel findings on the underlying mechanism are required. A growing body of evidence suggests that an altered energy metabolism is closely related to atrial arrhythmogenesis, and this finding engenders novel insights into the pathogenesis of the pathophysiology of AF. In this review, we provide comprehensive information on the mechanistic insights into the cardiac energy metabolic changes, altered substrate oxidation rates, and mitochondrial dysfunctions involved in atrial arrhythmogenesis, and suggest a promising advanced new therapeutic approach to treat patients with AF.

## 1. Introduction

Atrial fibrillation (AF) is the most common type of sustained arrhythmia in diabetes mellitus (DM). Its morbidity and mortality rates are high, and its prevalence will increase as the population ages. Despite expanding knowledge on the pathophysiological mechanisms of AF in DM, current pharmacological interventions remain unsatisfactory; thus, an innovative mechanistic understanding is necessary. Proteomic, metabolomic, and lipidomic analyses have indicated that substantial metabolic changes are involved in human and experimental AF pathophysiology; this finding has engendered novel insights into the pathogenesis of AF [1,2,3]. A proteomic analysis demonstrated that the differences in protein expression between the left atrial appendage tissues of patients with AF and the sinus rhythm (SR) of healthy individuals were closely related to energy metabolism [3]. The primary enzymes responsible for glycolysis and the tricarboxylic acid (TCA) cycle were downregulated, and long-chain fatty acids (FAs) were upregulated in patients with AF [2,3]. Moreover, a proteomic analysis demonstrated that cholesterol metabolism and lipid metabolism or pathways, such as lipid binding and transfer, were dysregulated in patients with AF [2]. The downregulation of the proteins responsible for glucose metabolism may contribute to the development of AF after cardiac surgery [4]. Patients with persistent AF exhibited an increased number of enzymes and metabolites involved in the metabolism of ketone bodies [5]. These findings suggest that cardiac metabolism plays a vital role in atrial arrhythmogenesis. 

DM has been reported to induce cardiac dysfunction (DM cardiomyopathy (DCM)) due to mitochondrial dysfunction [6]. Individuals with DCM are associated with an increasing risk of AF. Growing evidence has shown that several molecular mechanisms, including hyperglycemia, inflammation, oxidative stress, connexin remodeling, lipotoxicity, and mitochondrial dysfunction, induce atrial structural and electrical remodeling in DM [7,8,9]. Our previous study has shown an increasing mitochondrial oxidative stress in DM cardiomyocytes [10]. Different antidiabetic agents have variable impacts on the risk of AF in DCM [11], suggesting that the different strategy for glycemic control with dissimilar energy utilization may play a vital role in the pathogenesis of AF in DCM. Accordingly, exploring the cross-talk between cardiac energy and atrial remodeling is key to understanding the pathophysiology of AF, and cardiac metabolism is expected to be a potential target for the treatment of AF.

## 2. Energy Metabolism in the Heart

The heart consumes much energy in the form of adenosine triphosphate (ATP) that is constantly refilled through oxidative phosphorylation (OXPHOS) in mitochondria and glycolysis in the cytosol [12], which are the primary energy generators [13]. Under physiological conditions, mitochondrial OXPHOS contributes to approximately 95% of the myocardial ATP supply, and glycolysis provides the remaining 5% [14]. FAs and carbohydrates are two primary energy substrates of the heart, and ketone bodies and amino acids are minor sources of energy (Figure 1) [15,16]. FAs enter the cardiac myocytes through protein transporters on the cell membrane, which involves CD36/FAT and fatty acid-binding proteins (FABPs) [15]. CoA is added to the FAs by fatty-acyl-CoA synthetase (FACS) in the cell, enabling long-chain FAs to enter the mitochondria. Medium-chain FAs pass through the mitochondrial membrane without prior activation of acyl-CoA, but the mitochondrial membrane is resistant to long-chain FAs [17]. Carnitine palmitoyl transferase 1 (CPT-1) converts the long-chain fatty-acyl-CoA to an acylcarnitine in the cytosol, enabling the FAs to enter the mitochondria [15]. Acylcarnitine is further transported by carnitine translocase across the inner mitochondrial membrane.

Subsequently, long-chain fatty acylcarnitine is regenerated back to a fatty-acyl-CoA in the mitochondria by CPT-2, then enters repeated cycles of FA β-oxidation, which involve four enzymes that progressively reduce acyl-CoA, produce acetyl-CoA, and reduce equivalent nicotinamide adenine dinucleotide (NADH) and flavin adenine dinucleotide (FADH_2_) in each cycle [15,18]. The transport of long-chain acyl-CoA is dependent on the carnitine acyltransferase membrane shuttle because it cannot penetrate the mitochondrial membrane [19]. Medium-chain FAs, however, do not require the carnitine acyltransferase membrane shuttle to enter the mitochondria. Mitochondrial FAs uptake and, following FAs β-oxidation, are regulated by numerous factors, such as Malonyl-CoA, which is an endogenous inhibitor of CPT-1, glucose oxidation rate, and pyruvate dehydrogenase (PDH) activity level [18].

Glucose is another main source of energy for the heart, and its metabolism involves glucose uptake, glycolysis, and glucose oxidation in the mitochondrial matrix [20]. Glucose uptake into cardiac myocytes is executed by glucose transporters (GLUT1/4) that subsequently undergo glycolysis in the cytosol to produce ATP and pyruvate [19]. Pyruvate, the end-product of glycolysis, traverses the mitochondrial outer membrane through a voltage-dependent anion channel and subsequently crosses the inner mitochondrial membrane by the mitochondrial pyruvate carrier (MPC). In the mitochondrial matrix, pyruvate can be converted to acetyl-CoA by PDH, a rate-limiting enzyme for glucose oxidation [21]. PDH activity depends on its phosphorylation level; PDH is inhibited by PDH-kinase and activated by PDH-phosphatase [22]. 

In the mitochondrial matrix, acetyl-CoA derived from glycose oxidation and FA β-oxidation enters the Krebs cycle to produce NADH and FADH_2_, which donate their electrons to the oxidative phosphorylation process in the electron transport chain (ETC) to generate a proton gradient required for OXPHOS and ATP synthesis [23,24]. An increased FA oxidation rate elevates the levels of intracellular acetyl-CoA and citrate, which inhibit the activities of the enzymes involved in glucose metabolism. The increased glucose oxidation rate that suppresses FA oxidation is known as the Randle Cycle [25]. PDH is inhibited by NADH or acetyl-CoA produced from FA oxidation, and NADH or acetyl-CoA released from glucose oxidation inhibits the FA oxidation enzymes’ activities [20]. Evidence suggests that metabolic remodeling precedes electrophysiological, contractile, and structural remodeling in AF [26].

Ketone bodies are increasingly recognized as a vital energy substrate for the heart [27]. In a healthy heart, the oxidation of ketone bodies contribute to 10–15% of overall cardiac ATP production [28]. Ketone bodies are produced in the liver from acetyl-CoA derived from FA oxidation in the mitochondrial matrix when the glucose concentration is low or after prolonged starvation or intense exercise [1,27]. β-hydroxybutyrate (βOHB), acetone, and acetoacetate (AcAc) are the primary ketone bodies that can serve as additional energy sources for the heart. When ketone bodies are produced by the liver, they are released into the bloodstream to the heart for further oxidation in the mitochondria. βOHB is taken up by cardiomyocytes and converted into AcAc by βOHB dehydrogenase in the mitochondrial matrix. AcAc is further converted into acetoacetyl-CoA by succinyl-CoA:3-ketoacid CoA transferase (SCOT), the rate-limiting enzyme of ketone oxidation [29]. Mitochondrial thiolase then hydrolyzes acetoacetyl-CoA into two acetyl-CoA molecules that enter the TCA cycle to produce reducing equivalents for ETC to generate ATP [30]. Compared with other organs, the protein amount for SCOT is the most abundant in the heart, highlighting its large ketolytic capacity [31,32]. Research has demonstrated that an increased circulating level of ketone bodies and upregulated ketone body oxidation play a vital role in cardiovascular diseases [28,29,33,34,35]. 

### 2.1. Mitochondrial Dysfunction in DM Cardiomyopathy

Development and progression of DM cardiomyopathy have been associated with mitochondrial structural and bioenergetics dysfunction [36]. Increased plasma FFA levels and the higher availability supply of FAs exceed mitochondrial FAO capacity, result in the accumulation of FAO intermediates and lipotoxicity in the heart, leading to oxidative stress, calcium dysregulation, the uncoupling of OXPHOS, and mitochondrial dysfunction [37,38]. Metabolically, the diabetic heart relies on FFA as the main energy substrate for oxidative metabolism and is characterized by downregulated glycolysis and upregulated FAO rates [39]. In early, fructose-induced DM hearts, PDH activity was decreased and the FAO pathway was increased along with increased cellular and mitochondrial FA uptake, as well as an increase in β-oxidation enzyme activities, despite reduced mitochondrial mass and long-chain acyl-CoA dehydrogenase activity [40], indicating altered energy metabolism in the heart abundantly relies on FA utilization. Diabetic hearts augmented the expressions and activities of PPARα and PGC1α to cope with excess FA availability, leading to the activation of genes involved in the FA metabolism, increased PDK4 protein levels, and the downregulation of glycose oxidation by decreasing PDH activity [41,42,43]. Moreover, diabetic cardiomyopathy hearts exhibited excess superoxide levels, significant reduction in mitochondria size and mitochondrial disorganization, and a significant increase in complex I-V protein abundance, along with reduced mitochondrial respiration [44,45]. These findings suggest that the excess influx of reducing equivalents of the TCA cycle may enter the ETC as a compensatory mechanism, inducing higher proton leak and uncoupled oxidative phosphorylation [46,47]. Similarly, DM patients with AF had a greater level of myocardial triglycerides, impaired complexes I, II, and IV activities, increased protein oxidation, and reduced maximal FA oxidation capacity in atrial appendages than nondiabetic patients [48,49,50], implying an increased mitochondrial ROS generation and uncoupled respiration in the diabetic atrium.

Mitochondrial dysfunction in HF with preserved ejection fraction (HFpEF): In the community, HF with preserved ejection fraction (HFpEF) accounts for approximately half of the global HF cases, and the prevalence of HF is increasing in the aging population [51]. AFs were among the three most common precipitating factors for hospitalizations and the most common comorbidities in those patients with HFpEF [52]. Main risk factors for HFpEF are obesity, metabolic syndrome, and diabetes [53]. Epicardial adipose tissue accumulation play an important role in the development of HFpEF, and HFpEF patients with AF and/or DM had higher epicardial fat [54]. Myocardial gene expression analysis showed that uniquely upregulated pathways in HFpEF are mitochondrial ATP synthesis and OXPHOS, and these pathways correlate with the differences in obesity degree between HFpEF and control patients [55], reflecting higher energy demand for those patients. Additionally, peripheral blood biomarkers in patients with HFpEF have shown increases of mitochondrial superoxide production and mitochondrial mass, decreased LDHB expressions, and higher lactate level [56]. Growing evidence also indicates that mitochondrial metabolism and its metabolic flexibility appears to play a major role in the pathogenesis of HFpEF, which have been extensively discussed in recent review papers [57,58]. 

### 2.2. Mitochondrial Dysfunction and ATP Deficiency

Mitochondria play a crucial role in ATP production, calcium dysregulation, energy metabolism, cardiac oxidative stress, and inflammation, which are critical to AF pathophysiology [59,60,61]. AF is commonly associated with mitochondrial dysfunction, indicated by increased reactive oxygen species (ROS) production, decreased ATP synthesis and consumption, and bioenergetic changes in cardiomyocytes [61,62]. Diminished preoperative mitochondrial respiration, downregulation of OXPHOS, and increased sensitivity to calcium-induced opening of mitochondrial permeability transition pores were highly associated with postoperative AF [63]. Short-term tachypacing progressively induced ATP deficiency, mitochondrial membrane depolarization, and impaired mitochondrial Ca^2+^ handling and respiration in cardiomyocytes [64]. 

ATP levels in the left atrial appendage of patients with paroxysmal AF were considerably higher than the levels of patients with SR; however, levels were lower in patients with longstanding persistent AF [64]. In stretch-induced AF, the PCr/ATP ratio was higher because of both a PCr increase and ATP production decrease in atrial tissue [65]. Short-term tachypacing significantly activated F0F1-ATPase activity without changing the mitochondrial biogenesis in sheep atria [66]. Consistent with these findings, the atrial tissue of patients with AF exhibited increased oligomycin A-sensitive ATPase activity and reduced ETC activity [67]. Our previous study determined that short-term tachypacing increased ATP and ADP production and enhanced ETC complex II activity, suggesting that heart rhythm is a key variable affecting atrial metabolism [68]. However, long-term tachypacing induced a considerable decrease in ATP production, F0F1-ATPase activity with downregulated ETC activity, and mitochondrial biogenesis [69]. These findings suggest that in paroxysmal AF or short-term tachypacing, compensatory mechanisms are initiated to match the ATP demand with a high rate of AF. This is later exhausted when AF lasts for a long period, which indicates mitochondrial dysfunction. 

Dysfunctional mitochondria are the source of a large amount free radicals that oxidize numerous intracellular targets, leading to an increase of ROS and arrhythmogenesis. AF is more common especially in an older population. It has been well recognized that central to the aging process of the heart is the overproduction of ROS that cause cellular oxidative damage, and the accumulation of this damage leads to the energetic dysfunction in aging [70]. The rise in obesity, trends in dietary patterns, and a chronic high-fat diet comes with the risk of lipotoxicity, which result in overproduction of ROS in mitochondria [71]. Mitochondria is the primary target of free radical production, and damaged mitochondrial DNA and dysfunctional mitochondria result in disrupted mitochondrial membrane potentials, reduced ATP production capacity, and mitochondrial respiration [72,73]. The subsequent aberrant mitochondrial signaling predisposes the myocardium to arrhythmias [74]. Moreover, under the metabolic stress condition, sensing the cellular energy deprivation, the sarcoK_ATP_ channels are triggered to open, leading to significantly shortened APD, which promotes the development of cardiac arrhythmia [75]. Low ATP levels affect the intracellular ionic current stability, decrease the efficiency of all energy-requiring enzymatic reactions, and impair contraction, relaxation, and ionic homeostasis in cardiomyocytes [76]. Low levels of ATP lead to the increased glycolysis and lactate synthesis that may be considered a mechanism similar to aerobic glycolysis, also known as the Warburg effect, in rapidly growing tumor cells [77].

Tightly controlled Ca^2+^ handling plays a main role in excitation–contraction coupling in cardiomyocytes [78]. Mitochondrial Ca^2+^ is critical to the regulation of mitochondrial ATP and ROS production, mitochondrial dynamics, and cell death initiation [79]. Excessive mitochondrial Ca^2+^ increases ROS production, opening the mitochondrial permeability transition pore, leading to apoptotic cell death and causing mitochondrial dysfunction and cellular contractile failure [80]. Mitochondrial Na^+^ is controlled by the Na^+^/H^+^ exchanger (NHE)-mediated Na^+^ uptake and mitochondrial NCX (mNCX)-mediated Na^+^ extrusion [81]. Under pathological conditions, an increase in cytosolic Na^+^ leads to a greater driving force for mNCX to extrude Ca^2+^ from the mitochondria, resulting in decreased mitochondrial Ca^2+^ content and cellular energy [82]. Potassium transport in mitochondria is controlled by the ATP-dependent mitoK_ATP_ channel, the Ca^2+^-dependent K_Ca_, and K^+^/H^+^ exchanger (KHE) [83]. Mitochondrial Ca^2+^ overload causes the activation of K_Ca_ and mitoK_ATP_ and the deactivation of KHE through mitochondrial membrane depolarization, the loss of the proton gradient, and decreased ATP pr FFA concentration obduction [84].

### 2.3. Fatty Acid Dysmetabolism in Atrial Arrhythmogenesis 

An impaired FA metabolism in the heart contributes to heart failure, hypertrophy, and arrhythmia, and increases in the plasma concentration of FA are associated with an increased risk of heart failure [85]. Accumulating evidence suggests that atrial arrhythmogenesis might be attributable to an impaired FA metabolism in the atria. Studies have indicated that higher plasma-free FAs among aging people are associated with a higher risk of AF, independent of age, sex, race, hypertension, and diabetes mellitus [86], and is a crucial independent predictor of AF-related stroke [87]. Shingu et al. have also demonstrated that patients with AF have higher serum-free FA levels and a greater expression of genes related to FA uptake and its transport in the atrium [88] than individuals without AF. This indicates that increased FA uptake and its oxidation in the atrium may contribute to the pathogenesis of AF. Moreover, free FA was postulated to inhibit the Na^+^/K^+^/ATPase pump, resulting in increased levels of intracellular sodium and calcium, which can induce arrhythmias. Similarly, a proteomic analysis of the left atrial appendage of patients with AF revealed that an increase in CD36 expression was not accompanied by an upregulation of proteins involved in FA uptake into mitochondria or subsequent β-oxidation in the mitochondria [89]. Elevated free FA and its increased uptake into the cell play a vital role in atrial arrhythmogenesis. The protein and mRNA expressions of CD36 and CPT-1, ATP production, and lipid accumulation substantially decreased in the atrial tissue of rabbits with rapid-pacing-induced AF [90]. Liu et al. demonstrated that the atrial tissues of patients with AF have lower expressions of CPT-1 and GLUT4 proteins than do patients with SR [91]. Fatty acid binding protein 3 (FABP3), involved in the uptake of FAs and their subsequent transport toward β-oxidation in the mitochondria, is downregulated in patients with new-onset AF, independent of age and atrial enlargement [92]. This indicates a decrease of FA oxidation and the transport of both FAs and glucose in the cell. Regarding new-onset AF, a recent study with a mean follow-up time of 23 years indicated that a dysfunction of the carnitine metabolism was substantially associated with an increased risk of AF [93]. Similarly, short-, medium-, and long-chain acylcarnitines were associated with new-onset AF [94], postoperative AF [63], and an increased risk of cardiovascular death and myocardial infarction [95]. Given that acylcarnitines play a vital role in mitochondrial FA oxidation, the increased concentration of acylcarnitine was deemed to be an indicator of impaired mitochondrial β-oxidation and metabolic stress [96]. Increased FA uptake is not always accompanied by a simultaneous increase in FA oxidation in the mitochondria during arrhythmogenesis [97], suggesting molecular mechanism that contributes to lipid accumulation in the atria. These findings may result from the fact that increased FA oxidation is an adaptive response to new-onset AF to match the sudden energy demand and failure of the adaptive response in energy metabolism, which contributes to longstanding AF. FFA concentration is significantly higher in DM subjects compared to control subjects [98]. Chronic high-fat diet consumption and downstream increased FAO, along with chronic low-grade inflammation in DM, can cause the trigger of an arrhythmia [99]. Excessive FFA is a potent inducer of ROS overproduction, resulting in lipotoxicity associated with calcium dysregulation, mitochondrial dysfunction, and cell death. FFA incubation increased ROS production and the treatment with inhibitors of the mitochondrial ETC decreased ROS production induced by FFA, indicating FFA oxidation delivers additional electrons to the ETC, which causes ROS overproduction [100,101]. It has long been known that CaMKII is activated by excess ROS through the oxidation of Met281/Met282 and regulates the major components of cardiomyocyte Ca^2+^ handling and plays important role in arrhythmogenesis [78]. In addition, excess FFA-induced ROS formation causes sarcoplasmic reticulum stress, leading to the sustained reduction of the ER Ca^2+^ load [102,103]. 

### 2.4. Glucose Dysmetabolism in Atrial Arrhythmogenesis

Studies on glucose uptake during AF have observed a decreased expression of SNAP-23, which is required for the cytosol membrane translocation of GLUT4, and a decreased membrane expression of GLUT4. These findings were associated with a reduction in glucose uptake and an increased membrane expression of FAT/CD36 and FA uptake in irregularly paced cardiomyocytes compared with normal or regularly paced cells [97]. Patients with permanent AF, however, had considerably downregulated GLUT4, mCPT-1, medium-chain acyl-CoA dehydrogenase, PPAR-α, Sirt-1, and PGC-1α levels, indicating a reduction in glucose uptake [91]. 

Rapid atrial pacing for 1 week resulted in increased lactate levels in the serum, a notable downregulation of GLUT4 and PDH, and enhanced pyruvate dehydrogenase kinase (PDK) 4 protein and mRNA expressions in rabbits with AF, indicating increased glycolysis but reduced glucose oxidation in their hearts [90,91]. The expression of proteins responsible for glycolysis and pyruvate oxidation, such as enolase, PDH, and aldose, were downregulated in the atrial tissue of postoperative patients with AF [4]. The protein expression and activity of PDK-1 and PDK-4, which inactivate the PDH complex in the mitochondria, were increased and accompanied by a significant downregulation of PDH in canine models of paroxysmal AF [104]. Additionally, high lactic acid content and increased lactate dehydrogenase A (LDHA) were detected in canine LA in rapid atrial-pacing-induced paroxysmal AF [104]. These results suggest that the impaired coupling of glycolysis with pyruvate oxidation in the mitochondria following increased lactate accumulation in the cytosol may lead to dysregulated electrophysiological characteristics in atrial myocytes. This indicates the occurrence of aerobic glycolysis in AF. 

### 2.5. Ketone Body Metabolism and Pathogenesis of Atrial Arrhythmogenesis

Knowledge of the role of ketone metabolism in atrial arrhythmogenesis is fairly limited. Elevated concentrations of βOHB and increased ketogenic amino acids and glycine in cardiac tissue were noted in patients with persistent AF, which indicates the potential role of ketone bodies [5]. Both heart failure and cardiac hypertrophy play a critical role in atrial arrhythmogenesis through increased atrial fibrosis, atrial electrical remodeling, and mechanoelectrical feedback. In patients with advanced human heart failure, an increased abundance of ketogenic β-hydroxybutyryl-CoA, associated with increased myocardial utilization of βOHB, and a decreased concentration of myocardial lipid intermediates were observed [105]. A substantial increase in the expression of the gene encoding SCOT was also observed [105]. A study demonstrated that acute perfusion of βOHB increased myocardial ketone body oxidation rates without altering glucose or FA oxidation rates and was a primary fuel source of a healthy heart [106]. Additionally, elevated ketone oxidation rates markedly increased TCA cycle activity, producing many reducing equivalents and increasing myocardial oxygen consumption [106]. Culturing healthy adult rats’ myocytes with βOHB resulted in a considerable improvement in myocyte excitation–contraction coupling under hypoxic conditions, which was posited to be a beneficial adaptation for the heart during periods of metabolic dysregulation [107]. However, the chronic effects of βOHB treatment on cardiac function and electrical properties remain unclear [108].

### 2.6. AMPK Activation with Potential Antiatrial Arrhythmogenesis 

Adenosine monophosphate (AMP)-activated protein kinase (AMPK) is a primary upstream regulator of cell energy metabolism (Figure 2). A low energy status and elevated AMP/ATP or ADP/ATP ratio under metabolic stress activates AMPK through the phosphorylation of α-subunit Thr-172 through liver kinase B1 (LKB1) in the heart. Once activated, AMPK compensates for energy depletion by increasing energy production and suppressing energy consumption by regulating multiple molecular mechanisms, including cardiac electrophysiology, energy metabolism, mitochondrial homeostasis, and Ca^2+^ handling. In our previous study (Figure 3), we found that short-term tachypacing significantly increased pAMPK expression and mitochondrial Ca^2+^ content in HL-1 cardiomyocytes with and without *ZFHX3* KD (a common genetic variant with increasing risk of AF) [68]. The regulatory role of AMPK in energy metabolism during arrhythmogenesis is not fully clarified. In the atrium of canines exposed to short-term rapid pacing, the upregulation of total and pAMPK levels, increased FAT/CD36, decreased FA oxidation, and the downregulation of CPT-1 and ACC in the left atrium were observed [60]. However, in a chronic AF model, FAT/CD36, CPT-1, FA oxidation, and total and pAMPK were all substantially downregulated, and free fatty acids (FFA) and triglyceride accumulation and lipid deposition were increased in the atrium [61]. AMPK exerts protective effects on the development of arrhythmogenesis and prevents the progression of AF [109].

Extensive reviews have been published on the role of AMPK in cardiovascular disease [26,110,111]. Mice with cardiac-specific LKB1 knockout developed spontaneous AF, [112,113,114] accompanied by atrial structural, histopathological, and mitochondrial dysfunctional changes, including a reduction in all mitochondrial oxygen consumption parameters [112,114]. AMPK activation with metformin treatment notably reduced the incidence of spontaneous AF and improved the mitochondrial function, gap junction proteins, fibrosis, and ultrastructural changes in the atrium [114]. LKB1 knockout mice with inactivated AMPK developed progressive atrial enlargements with inflammation, increased fibrosis, apoptosis, necrosis, and a disrupted ultrastructure, which contribute to the genesis of AF [113]. 

Harada et al. provided evidence that AMPK activation under metabolic stress and paroxysmal AF conditions compensates for Ca^2+^ handling and cell contractility in atrial cardiomyocytes [115]. In their study, the atrial phosphorylated AMPK to the total AMPK ratio substantially increased in the left atrium cardiomyocytes with 2 Hz pacing with a glycolysis inhibitor. Patients with paroxysmal AF exhibited increased AMPK phosphorylation at the Thr172 site, but those with longstanding persistent AF exhibited a decrease [115]. Therefore, the initial AMPK activation is more likely to be adaptive, to be a compensatory response to energy deficiency during the onset of paroxysmal AF, and to maintain atrial functional integrity to protect individuals with paroxysmal AF from arrhythmia persistence. However, in the long term, the initial increased pAMPK in atrial myocytes may not meet the accumulating high energy demand of patients with chronic AF; thus, pAMPK levels tend to decrease, which contributes to the persistence of AF. [115,116].

### 2.7. Warburg Effect in Atrial Arrhythmogenesis

Under normoxic conditions, healthy cells metabolize glucose through glycolysis and then generate pyruvate for further oxidation in the mitochondrial matrix. Under hypoxic conditions, however, the mitochondrial oxidative metabolism is limited, and the pyruvate converts to lactate rather than glucose oxidation. This process was originally observed in cancer cells under a normoxic environment [117]. Energy production shifts from mitochondrial oxidative phosphorylation to aerobic glycolysis, a process known as the Warburg effect [77]. Aerobic glycolysis generates less ATP than mitochondrial oxidative phosphorylation, but aerobic glycolysis is faster and generates ATP more rapidly than mitochondrial oxidative phosphorylation [118,119]. Research has indicated that the Warburg effect not only occurs in tumor cells but also plays a vital role in the progress of nontumor diseases [119]. A primary determinant of the Warburg effect is increased expressions of PDK and LDHA, a higher lactic acid content, and upregulated hypoxia-inducible factor 1α (HIF-1α) associated with tissue ischemia in the myocardium [119]. 

Additionally, metabolic intermediates generated during aerobic glycolysis play a substantial role in cellular functions, including cell proliferation and metabolism [120]. During AF, the irregular high frequency excitation and contraction increase the energy demand and oxygen supply in the atrium, and AF is therefore strongly associated with cardiac hypoxia [50]. In patients with paroxysmal or short-lasting persistent AF, mitochondria increase ATP production through oxidative phosphorylation [64]. However, in the long term, diminished ATP levels affect the intracellular ionic current and energy-requiring enzymatic reaction and impair the contractions and ionic homeostasis of the cell [76]. Accordingly, low ATP production of mitochondrial oxidative phosphorylation leads to an increase of glycolysis and lactate production from pyruvate in the cytoplasm, which is similar to Warburg effect-related metabolic stress [76]. 

In a study, patients with AF exhibited an increased atrial lactate production, which was positively correlated with atrial structural remodeling, as reflected by severe oxidative stress injuries and apoptosis [121]. In a canine model, paroxysmal AF increased the lactic acid content and the expressions and activities of PDK-1, PDK-4, and LDHA and the decreased the expressions of PDH and citrate synthase and the AMP/ATP ratio in canine LA [104]. In another study, GLUT1 and GLUT4 levels and glycolytic intermediate metabolites were upregulated in the atrial tissue of animals with AF [122]. These findings indicate that the Warburg effect and its intermediates play a vital role in atrial arrhythmogenesis.

NLRP3 inflammasome is a crucial inflammatory signaling complex and its activation is frequently observed in paroxysmal or long-standing persistent AF patients [123,124]. Rises in mitochondrial Ca^2+^ increases mitochondrial ROS production leading to NLRP3 inflammasome activation [125]. Recently, the activation of NLRP3/CaMKII signaling has been identified in patients with postoperative AF and these molecular substrates sensitize cardiomyocytes to spontaneous Ca^2+^ leak, arrhythmogenic afterdepolarizations, and inflammatory mediators [126]. Moreover, cellular metabolic pathways such as glycolysis, FAO, and ketone body oxidation changes are associated with NLRP3 inflammasome activation [127,128,129]. Moon et al. provided molecular mechanisms by which cellular superoxide dependent activation of FAO is critical to NLRP3 inflammasome activation in primary human macrophages [130]. In this study, the deficiency of NADPH oxidase 4 (NOX4) resulted in reduced CPT1A expression, leading to suppressed FAO oxidation and less activation of NLRP3 in human and mouse macrophages [130], suggesting NOX4-mediated FAO upholds NLRP3 inflammasome activation. 

## 3. Targeting Cardiac Metabolism as an Upstream Treatment of AF 

Evidence from translational research has suggested that cardiac metabolism may be a novel target as an upstream treatment for AF. Novel agents or lifestyle modifications that have normalized or improved cardiac metabolism are deemed potential targets for the reduction of the AF burden or the prevention of AF progression. 

### 3.1. Activation of AMPK: A Potential Therapeutic Strategy

After a 13-year follow-up, the use of metformin, an AMPK activator, independently protected patients with diabetes from new-onset AF [131], and metformin use is associated with a lower risk of hospitalization for AF in patients with type 2 diabetes mellitus [132]. In in vitro studies, 4 Hz tachypaced HL-1 atrial cardiomyocyte increased oxidative stress and myolysis, and treatment with metformin reversed these abnormalities [131]. In a canine model, metformin significantly increased the AMPK expression in the rapid pacing group of canines with AF compared with the SR group. Metformin reduced the concentrations of FFA and triglyceride and lipid accumulation in the left atrial appendage and promoted β-oxidation of FA in AF models through the AMPK/PPAR-α/VLCAD pathway [116]. 

### 3.2. Ketogenic Diet or Ketone Administration on Risk of Arrhythmogenesis

Because of the increasing incidence of obesity and diabetes, both of which are associated with arrhythmogenesis, low carbohydrate–high fat ketogenic diets have gained popularity because of their ability to induce short-term weight loss without hunger [133]. The ketogenic diet improves various cardiometabolic risk factors, including obesity, hypertension, dyslipidemia, hyperglycemia, and insulin resistance, in the short-term [134]. The prospective cardioprotective effects of the ketogenic diet, in rats, may be attributable to an increase in the number of mitochondria [135], the transcriptional upregulation of several OXPHOS genes [136], or the decrease in oxidative stress and mitochondrial DNA damage, which enhances the mitochondrial antioxidant status [137]. Additionally, the ketogenic diet for 3 weeks reduces cellular ROS and stimulates the cellular endogenous antioxidant system in hippocampal mitochondria and liver tissue from KD-fed rats [138]. There is an increase of ETC activity or mitochondrial biogenesis through SIRT1 activation in hippocampus from KD-fed rats, strongly suggesting neuroprotective effects of KD [139,140]. Moreover, KD may decrease systemic inflammation through multiple mechanisms [141]. 

It remains controversial regarding the cardiovascular benefits and safety of long-term use of ketogenic diets [142]. In a large, prospective, cohort study with a long-term follow-up of >20 years, low carbohydrate intake was associated with a higher risk of AF incidence, independent of other well-known risk factors for AF [143,144]. In a population-based cohort study with an average 15.7-year follow-up, low carbohydrate–high protein intake was associated with a considerable increased incidence of cardiovascular diseases, such as ischemic heart disease and ischemic stroke [145]. Ketonemia induced by a 2-week ketogenic diet in rats resulted in lower circulating insulin and myocardial glycogen store, impaired left ventricular function, and increased myocardial injuries following ischemia–reperfusion in the isolated heart [146]. Long-term ketogenic diets may potentially exert some systemic effects, such as metabolic acidosis [147,148,149], leading to arrhythmogenesis or a mechanoelectrical disturbance. Additionally, ketogenic diets are typically high in saturated fat and cholesterol, which may also induce systemic hyperlipidemia and intracellular lipid accumulation, which leads to increased fatty acid oxidation [150] and atrial arrhythmogenesis [87,151] due to enhanced ROS production [152,153,154]. Additionally, the decreased consumption of whole grains, fruits, vegetables, legumes, and fiber and increased consumption of animal products in the ketogenic diet may potentially increase body inflammation [155,156], which is a primary risk factor for AF. A clinical trial of 20 children on the ketogenic diet revealed a prolonged QT interval and cardiac chamber enlargement and dysfunction [157]. Substantial correlations between a prolonged QT interval, low-serum bicarbonate, and high ketone body level were noted [157,158]. These findings suggest that the ketogenic diet exerts many biological effects in addition to weight loss. However, long-term use of ketogenic diets to maintain endogenous ketosis raises several other concerns regarding the risks of arrhythmogenesis and chronic dietary intervention is not an ultimate solution. 

Ketosis would be reached through exogenous ketone supplementation, such as ketone salts and ketone esters. Of the exogenous ketone supplementations, the ketogenic effects of ketone esters are more prominent than ketone salts. Ketone salts administration results in a substantial sodium intake, which is detrimental for long-term use among patient with cardiovascular disease. As summarized in the Table 1, recent clinical trials have investigated the acute effects of exogenous ketone administration in HF patients or healthy individuals with promising findings. Short-term ketone administration as an acute treatment, both ketone ester and ketone salts induced dose-dependent ketosis. Exogenous ketone salt infusions in patients with HFrEF may improve cardiac function and LVEF by augmenting myocardial oxygen consumption without altering mechanoenergetic coupling [159]. 

Although the long-term effects of exogenous ketone in humans were not available, several animal studies have found that ketone administration may improve myocardial dysfunction in mice through multiple molecular mechanisms (Table 2). Chronic treatment of ketone esters in a mice HF model demonstrated that induced sustained ketonemia, reduced cardiac fibroblasts, and collagen deposition improved LV function and restored myocardial ATP production [33,163]. In tachypacing-induced canine HF, continued infusions of ketone salt treatment evidently reduced cardiac dilatation and improved LVEF, along with reduced myocardial glucose and its oxidation rates, without changing FFA oxidation [34]. Although limited data are available, these findings provide the evidence that exogenous ketone bodies have promising therapeutic potential for arrhythmogenesis. 

### 3.3. Ketone Body Modulation by Using SGLT2 Inhibitor 

Recent human and animal studies have demonstrated the potential cardioprotective effects of selective inhibitors of sodium–glucose cotransporter 2 (SGLT2) [166]. The effect of canagliflozin, an SGLT2 inhibitor, was assessed on atrial electrical and structural remodeling and oxidative stress states in a canine AF model [167]. The administration of canagliflozin induced mild hyperketonemia and suppressed electrophysiological changes, the degree of interstitial fibrosis, and the amount of oxidative stress in the treatment group (in comparisons with a 3-week rapid pacing control group) [167]. Another SGLT2 inhibitor, dapagliflozin, has been reported to reduce the incidence of AF in high-risk populations with and without diabetes mellitus [168,169,170]. SGLT2 inhibition with empagliflozin diminished cardiac hypertrophy; reduced interstitial fibrosis, myocardial oxidative stress, and mitochondrial DNA damage; stimulated mitochondrial biogenesis; normalized glucose and FAs metabolisms; and improved cardiac function in rats without diabetes following myocardial infarction. This was associated with increased amounts of plasma ketone bodies and expressions of myocardial ketone body transporters and BDH1 and SCOT enzymes, suggesting increased ketone body utilization [171]. These findings suggest the role of ketone body utilization in atrial arrhythmogenesis. 

SGLT2 inhibitors have a direct cardiac effect on isolated ventricular and atrial cardiomyocytes [172,173]. The NHE is abundantly expressed in human atrial and ventricular tissue, and atrial NHE-1 expression was substantially increased in patients with HF and AF. The SGLT2 inhibitors dapagliflozin, canagliflozin, and empagliflozin inhibited NHE activity and normalized cytosolic Na^+^ and Ca^+^ concentrations [172]. Our previous study investigated the effect of long-term treatment of empagliflozin on Ca^2+^ dysregulation, the late Na⁺ channel, and NHE currents in ventricular myocytes from diabetic rats [10]. We found that empagliflozin-treated diabetic rats had lesser intracellular Na^+^ levels, late Na⁺ channel, and NHE currents than those without empagliflozin treatment (Figure 4) [10]. Higher cytoplasmic sodium levels may adversely affect energy supply and demand matching and can induce an increase in mitochondrial ROS [174]. However, whether SGLT2 regulates cardiac sodium and calcium homeostasis through its effects on ketone body utilization is unclear.

## 4. Conclusions

Substrate metabolic dysregulation plays a vital role in the pathogenesis of atrial arrhythmogenesis in DM. AF in DM is associated with multiple proteomic, metabolomic, and lipidomic remodeling. Targeting the atrial energy and electrical connection is a novel therapeutic approach to decrease cardiac metabolic distress, limit the energy demand, and slow the progression of AF in DM. 

## Figures and Tables

**Figure 1 cells-11-02915-f001:**
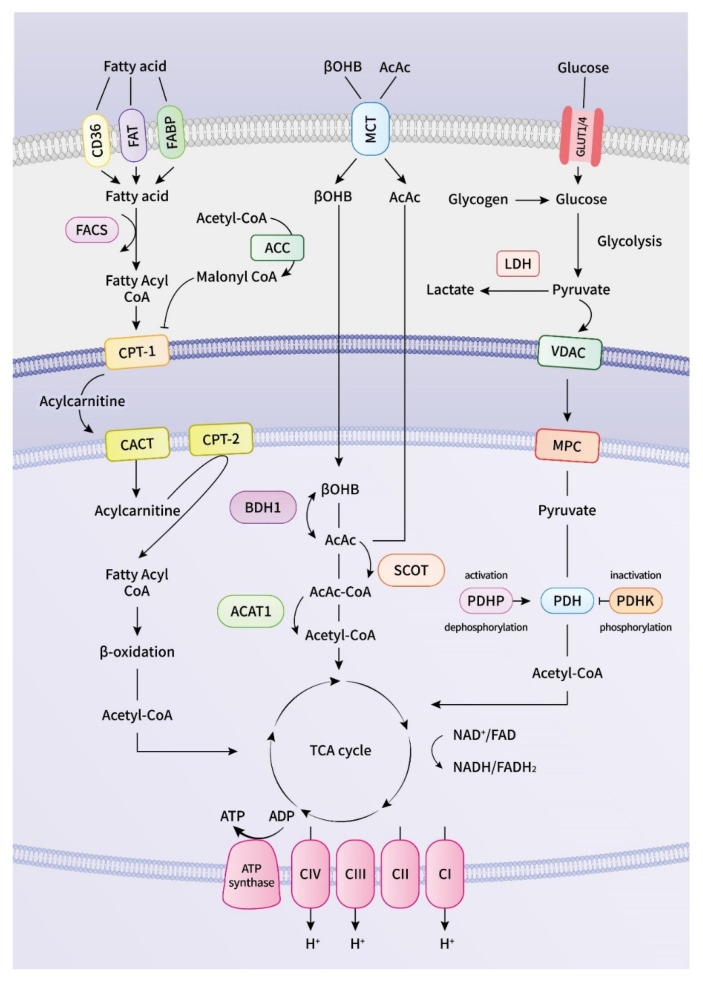
**Schematic illustration of myocardial energy metabolism.** In the mitochondria, acetyl-CoA generated from fatty acid, glucose, and ketone bodies oxidation enters the tricarboxylic acid (TCA) cycle to produce NADN and FADH_2_, which donate their electrons to the mitochondrial complexes of electron transport chain to generate a proton gradient required for OXPHOS and ATP synthesis. AcAc, acetoacetate; ACAT, acetoacetyl CoA thiolase; ACC, acetyl CoA carboxylase; BDH, β-hydroxybutyrate dehydrogenase; βOHB, β-hydroxybutyrate; CACT, carnitine–acylcarnitine translocase; CPT, carnitine palmitoyl transferase; FABP, fatty acid-binding protein; FACS, fatty-acyl-CoA synthetase; FADH, reduced flavin adenine dinucleotide; FAT, fatty acid transport; GLUT, glucose transporter; LDH, lactate dehydrogenase; MCT, monocarboxylic acid transporter; MPC, mitochondrial pyruvate carrier; NADH, reduced nicotinamide adenine dinucleotide; PDH, pyruvate dehydrogenase; PDHP, PDH phosphatase; PDHK, PDH kinase; SCOT, succinyl-CoA:3-ketoacid CoA transferase; VDAC, voltage-dependent anion channels.

**Figure 2 cells-11-02915-f002:**
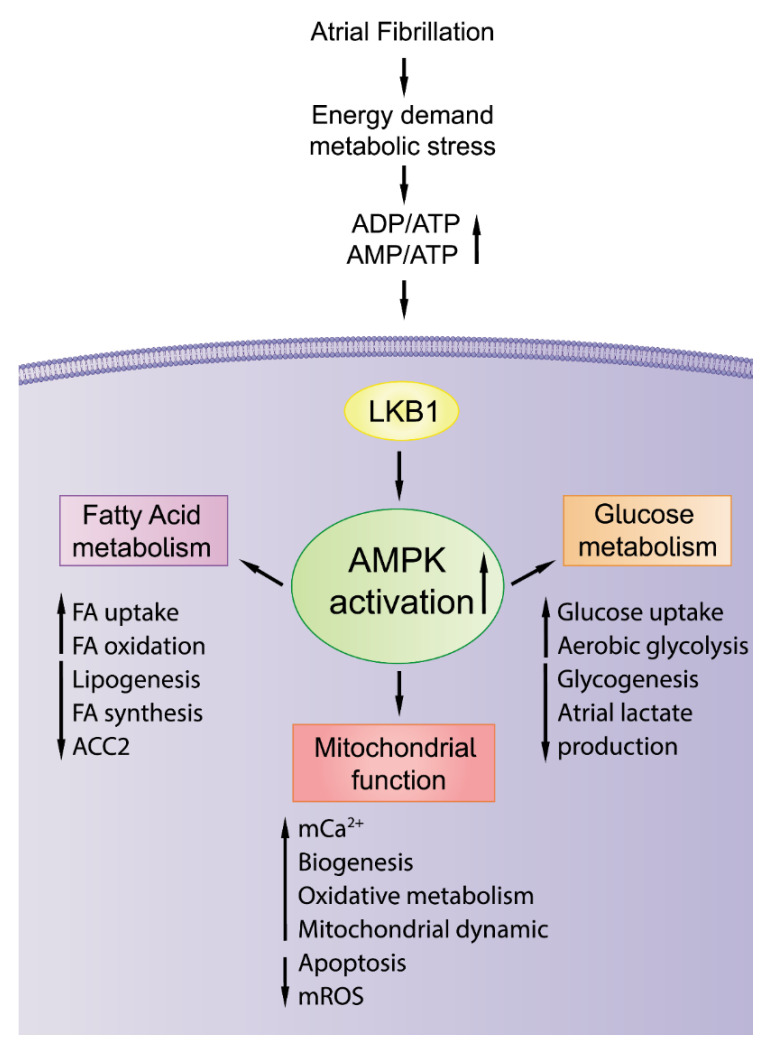
**Schematic illustration of the role of AMPK activation in****antiatrial arrhythmogenesis.** Atrial fibrillation enhances energy demand and induces metabolic stress. Metabolic stress causes an increase in ADP/ATP and AMP/ATP ratio. Elevated AMP/ATP or ADP/ATP ratio under metabolic stress activates AMP-activated protein kinase (AMPK) through the phosphorylation by LKB1 in cardiomyocytes. AMPK activation enhances energy production and limits energy consumption through FA and glucose metabolism and mitochondrial homeostasis. ACC, acetyl CoA carboxylase; FA, fatty acid; LKB1, liver kinase B1; mROS, mitochondrial reactive oxygen species.

**Figure 3 cells-11-02915-f003:**
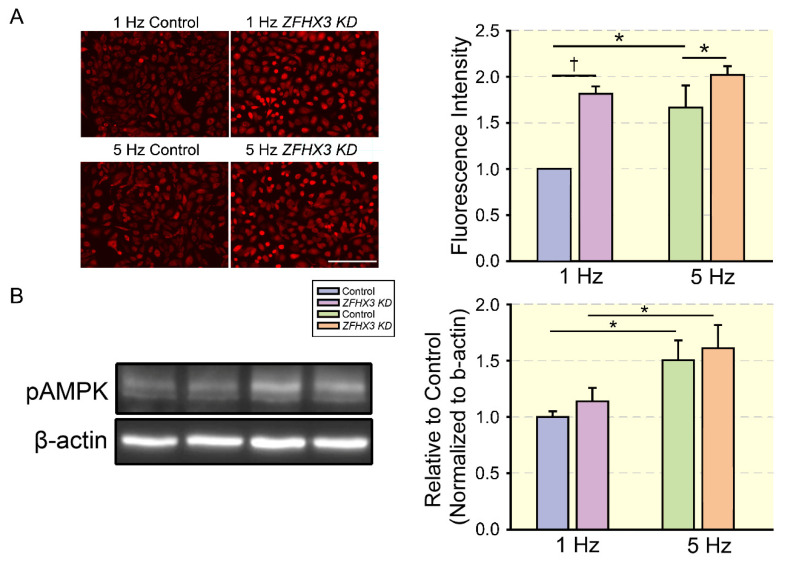
**Effects of the tachypacing on pAMPK protein expression and mCa^2+^ content**. (**A**) Representative fluorescence and average data of microscopic images of control and *ZFHX3* KD HL-1 cells under 1 and 5 Hz pacing (*n* = 3 experiments per group). (**B**) Representative Western blot and quantified data revealed tachypacing caused a significant increase in pAMPK protein expression in control and *ZFHX3* KD cells (*n* = 5 experiments per group). * *p* < 0.05, † *p* < 0.01 (Reprinted/adapted with permission from Ref [68]. 2022, John Wiley & Sons.)

**Figure 4 cells-11-02915-f004:**
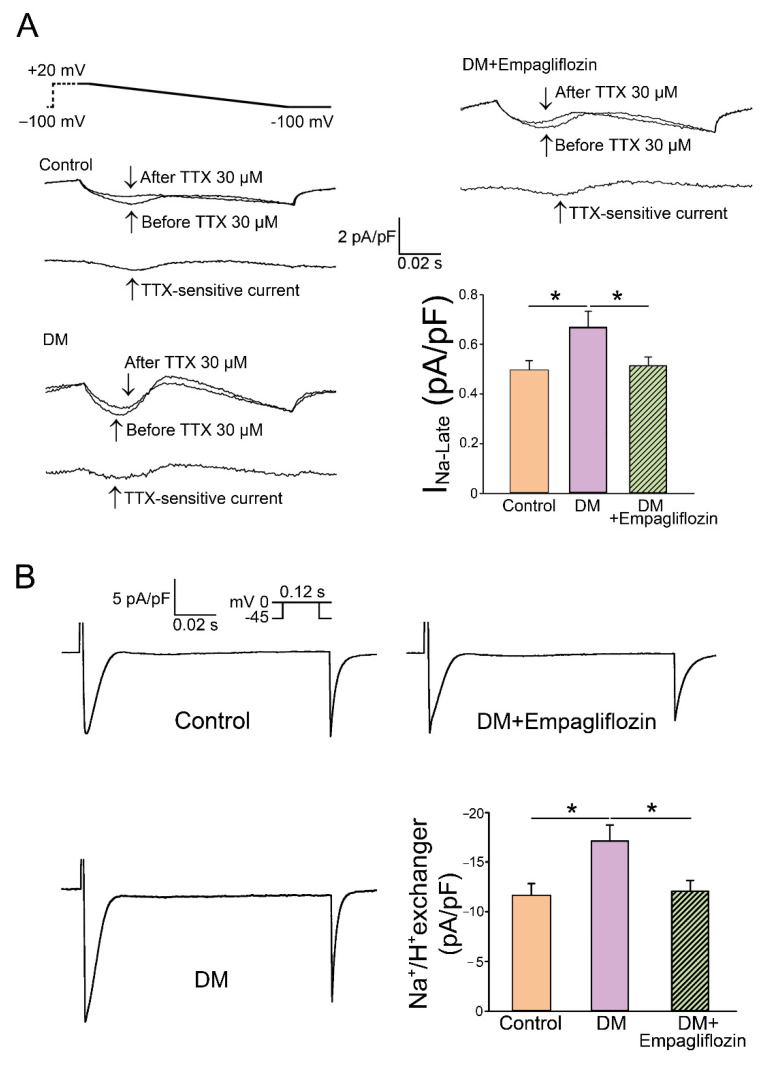
Modulation of late sodium current (I_Na-Late_) and sodium hydrogen (Na^+^/H^+^) exchanger current in empagliflozin-treated diabetic (DM) rats. (**A**) Representative tracing of current and average data of the I_Na-Late_ from control (*n* = 18 cells), DM (*n* = 15 cells), and empagliflozin-treated DM (DM+ empagliflozin, *n* = 16 cells) rat ventricular myocytes. (**B**) Representative tracings of current and average data of the Na^+^/H^+^ exchanger from control (*n* = 17 cells), DM (*n* = 18 cells), and DM + empagliflozin (*n* = 19 cells) rat ventricular myocytes. * *p* < 0.05. (Adapted from the published article by Lee et al. [10].)

**Table 1 cells-11-02915-t001:** Exogenous ketones administration on cardiac metabolism in healthy individuals or heart failure patients.

Study	Study Subjects	Intervention	Duration of Intervention	Main Outcomes
Luca Monzo et al. [160]	11 HFrEF patients 6 controls	Oral administration of 25 g of KE drink, 38% solution in water	80-min	12.9-fold increase of βOHB in peripheral venous Increased utilization of βOHB in HF group Fractional extraction of βOHB directly correlated with HF severity No relationship between enhanced βOHB extraction and FFA or glucose extraction Positive correlation between enhanced βOHB and lactate fractional extraction
Roni Nielsen et al. [159]	24 HFrEF patients 10 controls	7.5% Na-3-OHB infusion	3-h	Dose-dependent increase of circulating plasma βOHB levels Improved cardiac output by 40% and LVEF by 8% Improved myocardial oxygen consumption without worsening mechanoenergetic coupling
Lars C Gormsen et al. [161]	8 healthy individuals	7.5% Na-3-OHB infusion	390-min	Increased circulating βOHB levels Increased myocardial blood flow by 75% Circulating lactate levels were increased by 35% Halved myocardial glucose uptake No changes in FFA uptake and oxidative capacity
Donnie Cameron et al. [162]	28 healthy individuals	Oral administration of 25 g of KE drink	30-min	Mild ketosis Decreased blood glucose, lactate, and fatty acid concentrations

βOHB, beta-hydroxybutyrate; FFA, free fatty acids; HF, heart failure; HFrEF, heart failure with reduced ejection fraction; KE, ketone esters; LVEF, left ventricular ejection fraction; Na-3-OHB, Na-3-beta-hydroxybutyrate.

**Table 2 cells-11-02915-t002:** Translational researches of exogenous ketones administration on cardiac structural remodeling.

Study	Study Subjects	Intervention	Duration of Intervention	Main Outcomes
Shengen Liao et al. [164]	HFpEF mice induced by HFD + L-NAME fed	Intraperitoneal injections of βOHB at a dose of 10 mmol/kg	Once per week for 15 weeks	Mitigated diastolic dysfunction, reduced interstitial fibrosis, and cardiomyocyte sizes Prevented cardiomyocyte apoptosis, inflammation and oxidative stress
Yan Deng et al. [165]	HFpEF mice induced by HFD + DOCP injection	KE gavage of 1 mg/g body weight	Once per day for 30 days	Elevated myocardial βOHB levels Lowered mitochondrial hyperacetylation, suppressed NLRP3 inflammasome formation, elevated citrate synthase activity, inhibited fatty acid uptake
Shingo Takahara et al. [163]	HF mice induced by TAC surgery	20% KE (8.0 g ketones/kg/day) via drinking water or acute infusions of βOHB	2 weeks supplementation or 1 h infusions	Elevated blood βOHB levels, improved cardiac EF Reduced cardiac fibroblasts and collagen deposition Greater cardiac output No changes in blood glucose, fatty acid, and insulin levels
Salva R Yurista et al. [33]	HF mice induced by TAC/MI and in rats by Post-MI	Preventive KE-1 (10% *w*/*w*) or treatment KE-2 (3.4 ± 1 g per day) diet	KE-1 for 5 weeks KE-2 for 4 to 6 weeks	Induced sustained ketonemia Improved LVEF, reduced LV-ESV and LV-EDV Induction of genes involved in the myocardial uptake and oxidation of βOHB Restored myocardial ATP production.
Julie L Horton et al. [34]	Canine dilated cardiomyopathy induced by cardiac tachypacing	5 μmol/kg/min infusions of sodium-βOHB, 55 mL/day	14 days	2.5-fold higher plasma concentration of βOHB with increasing ketone body uptake Suppressed myocardial glucose uptake and its oxidation rates, reduced net lactate uptake Prevented myocardial oxygen consumption, increased mechanical efficiency by 30% Prevented changes in LVEDP, cardiac output, LV dilatation,

ATP, adenosine triphosphate; βOHB, β-hydroxybutyrate; DOCP, deoxycorticosterone pivalate; EDV, end-diastolic volume; EF, ejection fraction; ESV, end-systolic volume; HF, heart failure; HFD, high-fat diet; HFpEF, heart failure with preserved ejection fraction; KE, ketone esters; L-NAME, N^ω^ -nitrol-arginine methyl ester; LV, left ventricular; LVEDP, left ventricular end-diastolic pressure; LVEF, left ventricular ejection fraction; MI, myocardial infarction; NLRP3, NOD-like receptor protein 3; TAC, transverse aortic constriction.

## Data Availability

Not applicable.

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
