# Peer review of "Disturbed Cardiac Metabolism Triggers Atrial Arrhythmogenesis in Diabetes Mellitus: Energy Substrate Alternate as a Potential Therapeutic Intervention"

_cells, 2022, doi:10.3390/cells11182915_

Round 1
Reviewer 1 Report
This review manuscript had chosen an interesting topic. It was professionally written and organized. The citation of references is appropriate and comprehensive. The main conclusion and comments are useful to the clinical and scientific society, especially in atrial arrhythmogenesis field. This reviewer has the following comments and suggestions to improve the quality of this manuscript:
11) The author had categorized the total metabolism stress related to AF into sub-sessions based on metabolic substrates, which is good. However, since atrial fibrillation is a common comorbidity in patients with heart failure with preserved ejection fraction (HFpEF), it is highly recommended that the authors can integrate this part into an individual session, which is poorly covered or partially presented in other sessions.
22) The authors should pay special attention to the integrity of the reference cited in this manuscript. Some references’ information is not completed, e.g. [14], [17], [25], [34], [64], [75] the page numbers are not complete. Some of the references are duplicated, e,g. [94] = [95], [130] = [131]. The authors should keep the refence format consistent.
33) Page 11, line 390-393. The authors should clarify which of these effects by ketogenic diets are discovered in heart disease and which part is in the neuronal disease, otherwise it is ambiguous.
Reviewer 2 Report
In this narrative review, Dr. Lkhagva and colleagues explored the role of glucose and energy metabolism in atrial fibrillation. Overall, this is a nicely written review and this piece of work could be useful for future research in the field. However, some issues are identified and need to be addressed by the authors:
1) I believe that the authors need to explain more about the connection between (FA/ glucose/ ketone / etc) metabolism and AF substrates, drivers and triggers. When I read the manuscript, I expected to discover the link between those metabolism disruption with the electrophysiological components of AF. For instance, how could mitochondrial dysfunction cause ion channel dysfunction, leading to AF? I see that there is one sentence regarding Na/K channel but this is not enough.
In general, the authors have to link the energy metabolism-related pathophysiology with the ones in this review article (PMID: 32188566) because I think calcium-handling dysregulation would be one of the main hub.
2) If I read the text, the authors often discussed data in the setting when the AF is already present and the cellular metabolism parameters are measured thereafter. For example, this sentence "Short-term tachypacing progressively induced ATP deficiency, mitochondrial membrane depolarization, and impaired mitochondrial Ca2+ handling and respiration in cardiomyocytes" and many more. So, I am confused. Is AF causing metabolism issue in the atrial cardiomyocyte or is energy metabolism disturbance causing AF? At the moment, the latter is scarcely discussed in the text and this part needs to be expanded to provide a more balanced story.
3) Regarding section 2.1, I am not really sure how it is related to the title (AF)? This manuscript is not about DCM. I think this part needs to be rephrased.
4) I think the discussion regarding NLRP3 inflammasome needs to be expanded considering the established connection between NOX-4-associated fatty acid oxidation with it (PMID: 27455510). NLRP3 is known to strongly associated with postoperative AF (PMID: 32762493).
Round 2
Reviewer 2 Report
Thanks for the response. I have no further comments.